# The Effect of Different Concentrations of Total Polyphenols from *Paulownia* Hybrid Leaves on Ruminal Fermentation, Methane Production and Microorganisms

**DOI:** 10.3390/ani11102843

**Published:** 2021-09-29

**Authors:** Julia Puchalska, Małgorzata Szumacher-Strabel, Amlan Kumar Patra, Sylwester Ślusarczyk, Min Gao, Daniel Petrič, Maria Nabzdyk, Adam Cieślak

**Affiliations:** 1Department of Animal Nutrition, Poznan University of Life Sciences, Wolyńska 33, 60-637 Poznan, Poland; puchalska00@icloud.com (J.P.); malgorzata.szumacher@up.poznan.pl (M.S.-S.); min.gao@up.poznan.pl (M.G.); 2Department of Animal Nutrition, West Bengal University of Animal and Fishery Sciences, 37 K. B. Sarani, Kolkata 700037, West Bengal, India; patra_amlan@yahoo.com; 3Department of Pharmaceutical Biology and Botany, Wrocław Medical University, 50-556 Wrocław, Poland; sylwester.slusarczyk@umed.wroc.pl; 4Institute of Animal Physiology, Centre of Biosciences of Slovak Academy of Sciences, Šoltésovej 4-6, 040-01 Košice, Slovakia; petric@saske.sk; 5Department of Animal Physiology, Biochemistry and Biostructure, Laboratory of Animal Anatomy, Poznan University of Life Sciences, Wojska Polskiego 71C, 60-625 Poznan, Poland; marianabzdyk@gmail.com

**Keywords:** ruminants, *Paulownia* Clon In Vitro 112, polyphenols, methanogens, ruminal fermentation

## Abstract

**Simple Summary:**

*Paulownia* hybrid leaves, a wood industry by-product, have been recognized not only as a valuable dietary ingredient for ruminants but also as a rich source of bioactive components such as polyphenolic compounds. Hence, this study was carried out to assess the effects of different concentrations of *Paulownia* hybrid leaves rich in polyphenols or their particular part on in vitro ruminal fermentation, methane production and microbial populations. Paulownia leaves with high (PLH), medium (PLM), and low levels of polyphenols (PLL) were used from different plantation areas. Lamina (PLLA) and twigs (PLT) of the leaves were also collected from the PLM plantation. Most of the basic nutrient content was similar to dehydrated alfalfa. The use of *Paulownia* leaves with high content of total polyphenols (PLH and PLLA) or high content of fiber (PLT) lowered ruminal methane production. Reduction of methane production was attributed to the lowering of methanogen populations in PLH and PLLA without affecting substrate degradability and volatile fatty acid concentrations, but in PLT, it was due to decreased in vitro degradability causing decreases in some ruminal parameters, mainly total volatile fatty acids.

**Abstract:**

This experiment was conducted to study the effects of different concentrations of polyphenols of *Paulownia* Clon In Vitro 112^®^ leaves or their particular parts on in vitro ruminal fermentation, methane production and microbial population. *Paulownia* leaves with high (PLH; 31.35 mg/g dry matter (DM)), medium (PLM; 26.94 mg/g DM), and low level of polyphenols (PLL; 11.90 mg/g DM) were used from three plantation areas. Lamina (PLLA; 33.63 mg/g DM) and twigs (PLT; 2.53 mg/g DM) of leaves were also collected from the PLM plantation. The chemical analyses of *Paulownia* leaves indicated that the content of the most basic nutrients (e.g., crude protein concentration of 185 g/kg of DM) were similar to dehydrated alfalfa. The in vitro results showed that the use of *Paulownia* leaves with the highest content of total polyphenols (PLH and PLLA) decreased methane production, methanogens numbers, and acetate to propionate ratio. In PLT, lowered methane production was followed by reduced substrate degradability and volatile fatty acid (VFA) concentration along with higher acetate to propionate ratio. Therefore, reduction of methane production in PLH and PLLA was attributed to the lowered methanogen population, whereas in PLT it was caused by decreased substrate degradability with the resultant of limited hydrogen availability to the methanogens.

## 1. Introduction

Dairy cow production results in a great deal of threats to environments. Increased greenhouse gas emission into the atmosphere, alterations of biological diversification of the ecosystems, and some nutrient pollutions are only some of them [1]. Generally, animal production causes 20% of global greenhouse gases emission, especially methane and carbon dioxide [2]. Among animals, cattle are the main methane producer with a daily production of 250–400 L. Consequently, there is a need for searching management techniques in animal nutrition, which will provide stability and durability of the environment [3]. The environmental impact of animal production may vary depending on such nutritional practice [4]. Hence, there is a growing interest in finding a solution which will result in a sustainable and environmentally friendly production method. Including alternative dietary components in cattle nutrition with the potential to mitigate greenhouse gas production and not competing with human diets are particularly desirable. In recent years, some potential alternative feed sources for ruminants are being intensively analyzed and introduced into practical nutrition [5]. Especially, the utilization of a by-product as an animal feed is greatly useful since a significant amount of waste biomass is produced annually from crops or wood processing [5,6]. Nevertheless, usage of such by-products is often limited due to the lack of knowledge on nutritional value and the effects on performance data [7].

One of the plants with a great potential to be implemented as an alternative dairy cattle dietary ingredient is *Paulownia*, a tree originating from China [8]. *Paulownia* hybrid leaves, a wood by-product, have been recognized as a potential dietary ingredient for dairy cows as well as a rich source of bioactive components such as phenolic compounds, saponins, and flavones [8,9]. Recently, numerous studies have focused on the presence of these bioactive substances and their positive impact on animal health. It has been shown that they have the potential to modulate ruminal microbial fermentation mitigating methane production [10,11]. However, it should be kept in mind that the content of bioactive components may differ among the same species grown under different conditions [12]. Research conducted by Huang et al. [9] showed that different levels of bioactive components in *Paulownia* leaves and *Paulownia* leaves silage, differing in polyphenolic content (25 vs. 47 mg/g dry matter (DM)), may have different effects on rumen fermentation, including methanogenesis. Thus, it seems interesting to verify how *Paulownia* leaves from plants grown at different plantations containing the same bioactive components but different concentration will affect the rumen microbial environment.

We hypothesized that the higher concentration of total polyphenols may limit the production of methane due to a decrease in the methanogen populations. The main aim of the study was to assess whether different concentrations of polyphenols from *Paulownia* hybrid leaves from different plantation areas or their particular parts could influence in vitro ruminal fermentation, methane production and microbial populations.

## 2. Materials and Methods

### 2.1. Plant Material and Analysis of Total Polyphenols and Basic Nutrient Components

Fresh *Paulownia* Clon In Vitro 112 leaves were collected from three different plantations in Poland: 1st in Tuchola (sandy soil, class IV), 2nd in Szczecinek (sandy soil, class III), and 3rd in Trzcianka (light permeable soil, class IV). Over the period March, April, and May 2019 (3 months before collection of plant materials for the study), the average temperature of 14 °C in all plantations, and average rainfall of 44 mm in Tuchola and Trzcianka, and 39 mm in Szczecinek were recorded. These plantation areas were selected on the basis of the previous research where different concentrations of total polyphenols in *Paulownia* leaves were detected (unpublished data). This study showed that leaves from plantation 1 were characterized with high content of polyphenols (PLH), leaves from plantation 2 were characterized with medium content of polyphenols (PLM), and leaves from the 3rd plantation area contained the lowest level of polyphenols (PLL). Lamina of leaves (PLLA) and twigs (PLT) were also collected from the second plantation with PLM. Plant samples were lyophilized using a freeze-drier (Christ Gamma 2–16 LSC, Martin Christ, Osterode am Harz, Germany), ground using a ZM 200 mill (Retsch, Düsseldorf, Germany; 1 mm sieve) and kept in dark until analysis.

The phytochemical analysis was focused on polyphenols content, since they were previously marked as the predominant bioactive compounds in *Paulownia* leaves [9]. Before analysis of total polyphenols in plants, materials were ground again to fine powder. Then, 100 mg of each material was extracted three times with 80% methanol for 60 min at 40 °C. The extract was evaporated to dryness, dissolved in 2 mL of Milli-Q water (acidified with 0.2% formic acid) and purified by Solid Phase Extraction (SPE) using Oasis HLB 3cc Vac Cartrige, 60 mg (Waters Corp., Milford, MA, USA). The cartridges were washed with 0.5% methanol to remove carbohydrates and then washed with 80% methanol to elute phenolics and saponins. Further, they were reevaporated to dryness and dissolved in 1 mL of 80% methanol (acidified with 0.2% formic acid), followed by centrifugation at 23,000× *g* for 5 min before undergoing the spectrometric analysis. All analyses were performed in triplicate for three independent samples and stored in a freezer at −20 °C.

Concentrations of polyphenols in plants materials were analyzed by Ultra-high-resolution mass spectrometry on a Dionex UltiMate 3000RS (Thermo Scientific, Darmstadt, Germany) system with a charged aerosol detector interfaced with a high-resolution quadrupole time-of-flight mass spectrometer (HR/Q-TOF/MS, Compact, Bruker Daltonik GmbH, Bremen, Germany). Phenolic compounds were chromatographed on an Kinetex C18 column (2.1 × 100 mm, 2.6 μm, Phenomenex, Torrance, CA, USA), with mobile phase A consisting of 0.1% (*v*/*v*) formic acid in water and mobile phase B consisting of 0.1% (*v*/*v*) FA in acetonitrile. A linear gradient from 7% to 50% phase B in phase A over 20 min was used to separate phenolic compounds with a short of 0.3 min calibration segment from 0 to 0.5 min. The flow rate was 0.3 mL/min and the column was held at 25 °C. Spectra were acquired in negative-ion mode over a mass range from *m*/*z* 100 to 1500 with 5 Hz frequency. Operating conditions of the electrospray ionization (ESI) ion source were as follows: capillary voltage 3 kV, dry gas flow 6 L/min, dry gas temperature 200 °C, nebulizer pressure 0.7 bar, collision radio frequency 700.0 Vpp, transfer time 100.0 μs, and pre pulse storage 7.0 μs. Ultrapure nitrogen was used as drying and nebulizing gas, and argon was used as collision gas. Collision energy was set automatically from 15 to 75 eV depending on the mass to charge number (*m*/*z*) of the fragmented ion. Acquired data were calibrated internally with sodium formate introduced to the ion source at the beginning of each separation via a 20 μL loop.

Data acquisition and processing of spectra were performed with Bruker DataAnalysis 4.3 software (Bruker Daltonik GmbH, Bremen, Germany). The concentrations of the phenolic derivatives in all plant samples were calculated as equivalents of chlorogenic acid (CAS 327-97-9, 3-caffeoylquinic acid). Stock solutions of chlorogenic acid were prepared in 80% methanol at concentrations of 3.1, and 4.1 mg/mL, respectively, and kept frozen until used. Calibration curves for these two compounds were constructed based on seven concentration points (from 3.9 to 500 µg/mL). Hyperoside was used to calculate the amount of flavonoids identified in the extract, as well as chlorogenic acid for phenolic acids. The sum of polyphenols was calculated based on concentration of both: flavonoids and phenolic acids. Bruker QuantAnalysis 4.3 software (Bruker Daltonik GmbH, Bremen, Germany) was used for this purpose. All analyses were performed in triplicate.

All plant samples were also analyzed based on AOAC methods [13] using the following method numbers: 934.01 for DM, 942.05 for ash, 962.09 for crude fiber (CF, using FOSS Tecator, Fibertec System), 976.05 for crude protein (CP, using a Kjel-Foss Automatic 16210 analyzer; Foss Electric, Hillerød, Denmark), and 973.18 for ether extract (EE, using Soxhlet System HT analyzer; FOSS, Hillerød, Denmark). The differences between DM and ash contents represented the organic matter (OM) content. Van Soest et al. [14] methods were used to determine the concentration of neutral detergent fiber (aNDF; determined with amylase and sodium sulfite and expressed without residual ash). The concentration of nonfiber carbohydrates (NFC) was calculated using the formula: NFC, g/kg DM = 1000 − (CP, g/kg DM + EE, g/kg DM + ash, g/kg DM + aNDF, g/kg DM) [15].

### 2.2. Batch Culture Experiments

Batch culture procedure was used following the method described by Huang et al. [9]. Briefly, the rumen fluid was collected from three ruminal cannulated Polish Holstein-Friesian dairy cows (body weight of 625 ± 25 kg; mean ± standard error) fed with following diets: maize silage (352 g/kg DM), alfalfa silage (172 g/kg DM), grass silage (75 g/kg DM), beet pulp (101 g/kg DM), brewer’s grain (63 g/kg DM), extracted rapeseed meal (38 g/kg DM), concentrate (189 g/kg DM), and mineral–vitamin mixture (10 g/kg DM). The study was approved by the Local Ethical Commission for Animal Research (permission no. 14/2019). From each cow, ruminal content (approximately 400 g) was collected from the top, bottom and middle parts of the rumen and squeezed through two layers of cheesecloth into a bottle (Schott North America Inc., Elmsford, NY, USA). The ruminal fluid was quickly transported to the laboratory keeping the bottle in a water bath maintained at 39 °C. Rumen fluid samples from three cows were pooled in an equal proportion and mixed with a buffer solution containing 292 mg K_2_HPO_4_, 240 mg KH_2_PO_4_, 480 mg (NH_4_)_2_SO_4_, 480 mg NaCl, 100 mg MgSO_4_.7H_2_O, 64 mg CaCl_2_.2H_2_O, 4 mg Na_2_CO_3_, and 600 mg cysteine hydrochloride per liter of double distilled water in the 1:4 ratio. Buffered rumen fluid (40 mL) was added to prewarmed serum flasks (125 mL) and incubated under CO_2_ at 39 °C. Finally, two batch culture sets were carried out: in the first set, PLH and PLL were tested, and in the second set, PLM, PLT, and PLLA were studied. In both experiments, the plant materials (PLH, PLL, PLM, PLT, and PLLA) were incubated as a sole substrate (400 mg in DM) to rule out other factors related to the diets and to obtain ruminal fermentation responses based on the substrate used. The experimental design used in the present study was established on the data published by Huang et al. [9]. In the first experiment, the materials with two different concentrations of total polyphenols were selected to find out the effect of concentration of total polyphenols. In the second experiment, different plant parts were selected to evaluate the effects of *Paulownia* leaves parts on ruminal fermentation. The incubation flasks (five flasks for each substrate and each run) sealed with aluminum caps and rubber stoppers were placed in an incubator for 24 h and recurrently mixed. Additionally, five bottles in each run containing buffered ruminal fluid only were also included as a blank control. Each experiment was repeated in three runs in three consecutive days. Finally, 35 bottles in the first batch culture and 50 bottles in the in second batch culture were analyzed.

### 2.3. Methane Production, Rumen Fermentation, and Microbial Population

The gas production, after 24 h of incubation, was determined by the displacement of syringe piston, which was previously attached to the serum flasks. Total gas production was calculated by subtracting gas produced in blank control from the test materials. Methane production was determined by sampling the headspace gas of the flasks by using a gastight syringe into gas chromatograph (SRI310, Alltech, PA, USA) that was supplied with a thermal conductivity detector and Carboxen—1000 column (mesh side 60/80, 15 FT × 1.8 INS.S, Supelco, Bellefonte, PA, USA). Nitrogen was used as a carrier gas at a constant flow rate of 30.0 mL min^−1^. Oven temperature was pre-programmed: 180 °C for 1.5 min at the beginning, then 20 °C min^−1^ to 220 °C. Afterwards, the gas samples were injected. Methane concentration was calculated by comparison of retention times with appropriate gases standards (Multax, Zielonki-Parcela, Poland) containing 1210 ppm of methane in nitrogen gas (99.99%). After termination of incubation, pH was measured immediately after opening bottles, using a pH meter type CP-104 (Elmetron, Zabrze, Poland). The ammonia concentration was analyzed by the colorimetric Nessler method previously described by Cieslak et al. [16]. The volatile fatty acid (VFA) concentrations were measured by gas chromatography (GC Varian CP 3380, Sugarland, TX, USA) equipped with a flame ionization detector and a capillary column (30 m × 0.25 mm; Agilent HP-Innowax, 19091N-133, Agilent Technologies, Santa Clara, CA, USA). The qualitative and quantitative identification of VFA peaks was performed using standards prepared by mixing individual VFA purchased from Fluka (Sigma Aldrich, MO, USA). In vitro dry matter degradability (IVDMD) was calculated based on loss of the incubated substrate DM weight after 24 h of incubation, corrected by the DM residue from the blank control. The protozoa counts were performed under a light microscope (Zeiss, type Primo Star no. 5, Jena, Germany) using buffered rumen fluid with a defined volume (100 µL) [16].

Methanogens and total bacterial populations were quantified by fluorescence in situ hybridization (FISH). Briefly, 50 μL of the ruminal fluid was diluted in phosphate-buffered saline (PBS) and pipetted onto 0.22 μm polycarbonate filters (Frisenette K02BP02500 Knebel, DK) and vacuumed (Vaccum KNF Vacuport-Neuberg, Trenton, NJ, USA). After vacuuming, the filters were transferred onto cellulose discs for dehydration in an ethanol series (50, 80, and 90%, 3 min each). For each sample, a series of identical filters were prepared to allow the determination of optimal hybridization. Hybridizations were carried out in 50 μL of hybridization buffer (0.9 M NaCl; 20 mM Tris/HCl, pH 7.2; 0.01% SDS) containing the oligonucleotides probes (S-D-Arch-0915-a-A-20). After hybridization, the filters were washed with a washing buffer (20 mM Tris/HCl, pH 7.2; 0.01% SDS; 5 mM EDTA) for 20 min at 48 °C. The filters were rinsed gently in distilled water, air-dried, and mounted on object glasses with VectaShield (Vector laboratories, number H-1000, Burlingame, CA) anti fading agent containing DAPI (4′,6-diamidino-2-phenylindole). To distinguish total count of bacteria (DAPI) from other methanogens in the rumen fluid, filters were left in 4 °C for 1 hour in the dark until visualized using a Microscope Axio Imager M2 (Carl Zeiss Iberia, Madrid, Spain).

### 2.4. Statistical Analysis

All data were analyzed using SAS statistical software (University Edition, version 9.4 SAS Institute Inc., Lane Cove, Australia). Chemical composition and contents of the main bioactive compounds of *Paulownia* leaves were tested by one-way analysis of variance (ANOVA), followed by Tukey’s post-hoc test. Batch culture experiments (24 h incubation) were performed in three repetitions and five replications (5 incubation vessels) for each of plant materials. Data were analyzed by ANOVA using the General Linear Model (GLM) procedures of SAS. No statistical influence of day was observed; therefore, this factor was deleted from the final model. Differences between treatments in experiment 1 and experiment 2 were tested using Tukey’s post-hoc test. Data were taken as statistically different when *p* < 0.05. All values are shown as group means with pooled standard errors of means.

## 3. Results

### 3.1. Basic Nutrient and Phytochemical Components Analysis

The content of DM, OM, CF, and NDF was greater in PLT in comparison to the other plant materials (*p* ≤ 0.013; Table 1). CP values ranged from 52 to 198 g/kg DM. The lowest CP content was noted in PLT (*p* ≤ 0.001). Crude fat and NFC dominated in PLLA with values of 32 and 294 g/kg DM, respectively.

The higher content (*p* < 0.001) of total polyphenols was recorded in PLLA, PLH, and PLM (33.63, 31.35, and 26.94 mg/g DM, respectively), while the lowest concentration was noted in PLL and PLT (11.9, and 2.53 mg/g DM, respectively; Table 2). Among the different polyphenolic compounds, acteoside was the highest concentration in all of the analyzed samples of *Paulownia* leaves. Among the plant materials, the highest concentration of acetoside was noted in PLLA (21.15 mg/g DM) compared to other groups. Among all investigated *Paulownia* leaves samples, majority of biologically active substances (only except apigenin 7) were significantly different (*p* ≤ 0.015). The 7- hydroxytomentoside, campneoside isomer and luteolin 4′,7-O-diglucuronide were dominant after acetoside.

### 3.2. Batch Culture Experiments

*Paulownia* leaves with higher concentration of bioactive compounds (both PLH in the first experiment and PLLA in the second one) affected the analyzed parameters of ruminal fluid and the microbial populations. The pH value was higher (*p* = 0.02) in PLH compared to PLL (Table 3, the first experiment). PLH caused a decrease (*p* ≤ 0.029) in ammonia concentration, total gas production, CH_4_ production, CH_4_ mmol/g DM, CH_4_/total gas, CH_4_/IVDDM (in vitro degraded dry matter), acetate to propionate (A/P) ratio, *Entodiniomorpha* population, total protozoa, and total methanogens. PLH significantly (*p* ≤ 0.006) increased the propionate, iso-valerate, and valerate proportions. PLT significantly increased (*p* ≤ 0.049) the pH value, acetate concentration and total methanogen numbers in the buffered ruminal fluid (Table 4, second experiment). Among the majority of the remaining ruminal fluid parameters (IVDMD, ammonia concentration, total gas production, CH_4_ production, CH_4_ mmol/g DM, CH_4_/total gas, CH_4_/IVDDM, and total VFA concentration), a significant decrease (*p* < 0.001) was noticed in PLT compared with the PLLA. In comparison to PLM, PLLA with higher concentration of polyphenols (33.63 mg/g DM) reduced total gas production, CH_4_ production, CH_4_ mmol/g DM, CH_4_/total gas, CH_4_/IVDDM. The methanogens content was the lowest in PLLA 1.86 × 10^7^/^mL^) in relation to PLM and PLT.

## 4. Discussion

A recent study demonstrated that *Paulownia* leaves are not only a valuable source of basic nutrients but may also reduce the methanogenesis in dairy cows’ rumen [9]. These findings encouraged us to delve into the impact of total polyphenols present in *Paulownia* leaves on the ruminal environment’s parameters and its mechanisms. To our knowledge, this is the first complete report on the relationship between different polyphenols concentrations in *Paulownia* Clon In Vitro 112 leaves and ruminal fermentation, methanogens population and methane production.

The *Paulownia* leaves are a good source of nutrients, especially CP, the most expensive nutrient in animal feeding. The average value of CP was 185 g/kg of DM that was relatively higher compared to the previous results—175 g/kg of DM [9]—however, existing divergence in mean of CP concentration cannot be a result of different variety variance. *Paulownia* Clon In Vitro 112 is a *Paulownia fortunei* and *Paulownia elongata* hybrid, while the materials used in Huang et al. study [9] was a *Paulownia tomentosa* and *Paulownia fortunei* hybrid. In Huang et al. study [9], the CP content ranged from 132 to 199 g/kg of DM, which suggests that the observed differences may result from natural differences in the CP content in the tested plant. Such an explanation is supported also by the lack of differences in the harvesting time (in both cases it was May; forming of a tree crown) or lack of differences in soil conditions. Without a doubt, high CP concentration in *Paulownia* leaves was close to the nutrient value of alfalfa the widely used in cattle feeding, rather than grass. This feature enforces *Paulownia* as an alternative feed component since there is an ongoing shortage of feeds of dairy cows not competing with human foods [18].

With regard to the nutrients composition, as a result of increased NDF concentration (average value for *Paulownia* leaves was 432 g/kg DM), difference was noted in the NFC content (average 232 g/kg DM). Concentration of NDF was slightly higher in *Paulownia* leaves than in dehydrated alfalfa (378 g/kg DM), but lower than in dehydrated grass (542 g/kg DM; INRA Tables; [19]) which confirms a closer affinity to alfalfa. Higher NDF values do not disqualify *Paulownia* leaves as a feed component, but it is worth keeping in mind that its concentration has a direct impact on feed intake [20].

Differences in concentration of total polyphenols were observed among *Paulownia* leaves materials (PLH, PLL, and PLM), as it was previously hypothesized. The concentration of polyphenols ranged from 11.90 mg/DM to 31.35 g/mg DM. Polyphenol concentration in PLM was comparable to the finding of Huang et al., [9] study (26.94 mg/g DM and 24.6 mg/g DM, respectively) with *Paulownia* leaves. The differences in *Paulownia* leaves materials may be a cause of divergent agronomic management [12] than soil condition. Acteoside was the dominant bioactive substance amongst analyzed polyphenol compounds. Lan and Yang [6] research claimed that this compound can reduce the activities of ruminal bacteria by lowering the hydrogen sink in the rumen, which was indirectly observed in our study as well. Such a phenomenon could be seen in the changes of acetic acid, propionic acid as well as acetate to propionate ratio in the PLH group. Nevertheless, there was no direct change in total bacterial population. However, acteoside may also have a direct impact on the methanogen population [21] which was also observed in our study with the reduction of both methanogen population (by 29 %) and CH_4_ per gram of DM (by 7%). These observations may indicate that the higher concentrations of polyphenols inhibit growth effect to a greater extent to some class of methanogens, while the activity of other types of methanogens may be increased reducing the efficiency of methane inhibition. In the present study, also the lack of changes in the number of total rumen’s bacteria that provide hydrogen for the methanogenesis process should be considered for final CH_4_ production. Another compound present in higher concentrations in PLL, PLH, and PLM was 7- hydroxytomentoside, and its concentration was within the range of 3.66 to 14.01 mg/g DM. Additionally, in other studies on *Paulownia* Clon In Vitro 112, 7- hydroxytomentoside was found as one of the dominant biologically active compounds [22]. 7- hydroxytomentoside as an iridoid compound mainly exhibits potent antiviral action [23], which may participate as a factor influencing changes in the ruminal fermentation including methanogenesis in the present study. On the other hand, protozoa numbers were reduced when PLH was used. Protozoa play an important role in the methanogenesis process by providing hydrogen to methanogens from fermentation of feeds and acting as symbionts [24]. Due to this, reducing protozoa numbers are usually linked with decreased methanogens population [25], which was observed also in the current study. Decreased methanogenesis in PLH was associated with increased proportion of propionate and reduced proportion of acetate resulting in lowered acetate to propionate ratio, which are usually noted in this type of experiments due to rechanneling of hydrogen from methanogenesis to propionate production via succinate pathway [26,27].

Increased pH and reduced ammonia concentration were observed in the PLH group. As a consequence, more suitable ruminal fluid environment was obtained. Increased pH by PLH may be a result of higher concentration of total polyphenols that could favor lactate-utilizing bacteria. For example, according to Balcells et al. [28], phenolic acids containing antimicrobial features may induce lactate to VFA metabolizing bacteria with simultaneous reduction of lactate concentration in the rumen.

Decreased ammonia concentration in the PLH may result from the ability of phenolic acids to complex with proteins, which makes them unavailable for rumen microorganisms and detrimental effect of phenolic compounds on proteolytic bacteria [9]. Additionally, reduced protozoal numbers in PLH compared with the PLL, which decreases microbial protein turnover, could be responsible for lowered ammonia concentration in PLH [26]. In this context, it is reported that CP in *Paulownia* leaves have high soluble fractions (0.51; [9]), which has a direct impact on ammonia concentration in the rumen fluid.

The second experiment was focused on using *Paulownia* leaves with medium concentration of phenolic compounds (26.94 mg/g DM, PLM), lamina of leaves (33.63 mg/g DM), and its twigs (2.53 mg/g DM). We showed that higher concentrations of polyphenolic compounds in PLLA have a similar impact on ruminal fermentation characteristics and microorganism populations. PLLA mitigated methane production by 16% compared with the PLM. Additionally, in PLT (with the lowest concentration of phenolic acids; 2.53 mg/g DM) a decrease in methane concentration was also observed (25 %), which was the resultant of increased NDF content in this part of leaves. Consequently, decreased methane production was a result of reduced digestibility of dry matter (by 15%) limiting hydrogen availability to methanogens. Increased NDF concentration induced a pH increase and IVDMD reduction with a direct impact on the content and VFA profile. Thereby, methane production reduction in the second experiment in PLLA and PLT groups was caused by different mechanisms. Moreover, PLT containing high fiber content decreased propionate proportion and acetate to propionate ratio, which are normally observed in high fiber forage diets compared with concentrate diets due to stimulation of fiber degrading bacteria that predominantly produce more acetate in comparison to propionate [26].

## 5. Conclusions

PLH, PLLA (contained the highest content of total polyphenols; 31.35 and 33.63 mg/g DM, respectively), and PLT (contained a high amount of NDF; 618 g/kg DM) affected methane and VFA profile. Incorporating *Paulownia* leaves with high total polyphenols or NDF content into the ruminants’ diet could improve the sustainability of ruminant production system, since it lowered methane production. Reduction of methanogenesis was obtained by lowering methanogens populations in PLH and PLLA whereas decreasing IVDMD caused a decreased efficiency in ruminal fermentation, mainly total VFA and reduced propionate proportion in PLT. This study also demonstrated that plantation areas affect polyphenolic compounds in *Paulownia* leaves, which subsequently exert ruminal fermentation and methanogenesis differently.

## Figures and Tables

**Table 1 animals-11-02843-t001:** Chemical composition of *Paulownia* leaves with lower concentration of bioactive components (PLL), *Paulownia* leaves with high concentration of bioactive components (PLH), *Paulownia* leaves with medium concentrations of bioactive components (PLM), *Paulownia* leaves lamina (PLLA) and *Paulownia* twigs (PLT).

Items	PLL	PLH	PLM	PLLA	PLT	SEM	*p*-Value
Dry matter (g/kg fresh matter)	245 ^b^	289 ^b^	307 ^ab^	284 ^b^	374 ^a^	11.1	0.001
Organic matter (g/kg DM)	878 ^b^	874 ^b^	873 ^b^	892 ^ab^	902 ^a^	3.37	0.013
Ash (g/kg DM)	121 ^a^	126 ^a^	127 ^a^	108 ^ab^	98 ^b^	3.37	0.013
Crude protein (g/kg DM)	194 ^a^	192 ^a^	168 ^a^	198 ^a^	52 ^b^	14.2	<0.001
Crude fiber (g/kg DM)	166 ^b^	162 ^b^	161 ^b^	151 ^b^	321 ^a^	15.1	<0.001
Crude fat (g/kg DM)	22 ^b^	24 ^b^	26 ^b^	32 ^a^	17 ^c^	1.12	<0.001
NFC (g/kg DM)	236 ^ab^	239 ^ab^	222 ^ab^	294 ^a^	215 ^b^	9.85	0.030
NDF (g/kg DM)	425 ^b^	414 ^b^	457 ^b^	368 ^c^	618 ^a^	22.1	<0.001

DM: dry matter; NFC: nonfiber carbohydrate: NDF: neutral detergent fiber without ash and assayed with α-amylase; SEM: standard error of the mean. Within each row, means with lower case superscripts (a–c) are significantly different at *p* < 0.05.

**Table 2 animals-11-02843-t002:** Contents of the main bioactive compounds (mg/g DM) identified in *Paulownia* leaves with lower concentration of bioactive components (PLL), *Paulownia* leaves with high concentration of bioactive components (PLH), *Paulownia* leaves with medium concentration of bioactive components (PLM), *Paulownia* leaves lamina (PLLA) and *Paulownia* twigs (PLT).

Peak	Rt (min)	λ_max_ (nm)	Molecular ion *m*/*z*[M-H] ^-^	MS * Main^_^ion	MS * Fragments	Formula	Identification	References	mg/g DM						
									PLL	PLH	PLM	PLLA	PLT	SEM	*p*-Value
1	1.5	248	345.1201	183.0664	165,137	C_15_H_22_O_9_	7-hydroxytomentoside	[17]	3.66 ^d^	13.25 ^c^	14.01 ^b^	15.09 ^a^	0.169 ^e^	1.66	<0.001
2	1.8	218,279	461.1672	315.1079	161,135	C_20_H_30_O_12_	Dicaffeoylacteoside	HMDB0039233	0.03 ^ab^	0.03 ^ab^	0.02 ^b^	0.02 ^b^	0.049 ^a^	0.003	0.006
3	2.7	326	487.1461	179.0332	161,135	C_21_H_28_O_13_	1-O-Caffeoyl-6-O-alpha-rhamnopyranosyl-beta-glycopyranoside		0.17 ^a^	0.14 ^b^	0.09 ^c^	0.1 ^c^	0.015 ^d^	0.02	<0.001
4	7.3		329.1245	167.0707	149,179	C_15_H_22_O_8_	3-(4-Hydroxyphenyl)-1,2-propanediol 4′-O-glucoside	HMDB0033082	0.04 ^c^	0.34 ^b^	0.36 ^ab^	0.43 ^a^	0.015 ^c^	0.05	<0.001
5	8.5	330	639.1935	161.0232	325,529,179,151	C_29_H_36_O_16_	Campneoside isomer		2.52 ^d^	6.63 ^a^	3.91 ^c^	5.31 ^b^	0.126 ^e^	0.61	<0.001
6	9.6	267,335	621.1112	351.0579	269,193	C_27_H_26_O_17_	Luteolin 4′,7-O-diglucuronide		2.1 ^c^	3.25 ^a^	2.24 ^bc^	2.81 ^b^	0.014 ^d^	0.31	<0.001
7	9.8	287,329	653.2109	161.0238	179,459,621	C_30_H_38_O_16_	Campneoside I	5315651 *	0.51 ^c^	1.2 ^a^	0.51 ^c^	0.84 ^b^	0.109 ^d^	0.10	<0.001
8	10.3	222,330	623.2017	461.1681	161,315	C_29_H_36_O_15_	Acteoside	HMDB0034843	5.94 ^c^	17.12 ^b^	17.36 ^b^	21.15 ^a^	1.751 ^d^	2.02	<0.001
9	10.9	222,330	623.201	461.1684	161,315	C_29_H_36_O_15_	Isoacteoside	HMDB0041025	0.25 ^b^	1.09 ^a^	1.14 ^a^	1.29 ^a^	0.125 ^b^	0.14	<0.001
10	11.3	285,327	591.1364	269.0462		C_27_H_28_O_15_	Apigenin 7-[rhamnosyl-(1->2)-galacturonide]	HMDB0038847	0.16	0.24	0.16	0.2	0.009	1.12	0.449
11	11.7	220,330	665.2111	461.1669	161,315	C_31_H_38_O_16_	Acetyl acteoside(tubuloside B)	9831166 *	0.09 ^b^	0.59 ^a^	0.57 ^a^	0.68 ^a^	0.178 ^b^	0.06	<0.001
12	12.9	220,329	651.2324	475.1813	175,193,160,329,313	C_31_H_40_O_15_	Epimeredinoside A	11399576 *	0.09 ^c^	0.71 ^a^	0.59 ^b^	0.8 ^a^	0.139 ^c^	0.08	<0.001
13	13.7	220,329	591.211	429.1736	161,285	C_29_H_36_O_13_	Didehydroxyacteoside		0.05 ^ab^	0.11 ^ab^	0.11 ^ab^	0.11 ^a^	0.014 ^b^	0.02	0.015
							Total polyphenols		11.9 ^b^	31.35 ^a^	26.94 ^a^	33.63 ^a^	2.53 ^b^	3.56	<0.001

*Paulownia* samples were analyzed in negative ionization mode. DM—Dry mass, *—based on PubChem information, HMDB—Human Metabolome Database identification, Values are means of three replicate. SEM—standard error of the mean. Total polyphenols—flavonoids and caffeoyl phenylethanoid glycosides. Within each row, means with lower case superscripts (a–c) are significantly different at *p* < 0.05.

**Table 3 animals-11-02843-t003:** The effect of *Paulownia* leaves with lower concentration of bioactive components (PLL) and *Paulownia* leaves with high concentration of bioactive components (PLH) on in vitro ruminal fermentation, methane production, and microbial populations (experiment 1).

Item	PLL	PLH	SEM	*p*-Value
pH	6.57	6.62	0.011	0.02
IVDMD	0.66	0.64	0.006	0.181
Ammonia, mM	9.92	8.38	0.289	0.005
Total gas, ml/g DM	301	290	1.227	<0.001
CH_4_, mmole	0.96	0.89	0.010	<0.001
CH_4,_ mmole/g DM	2.39	2.22	0.024	<0.001
CH_4_, mmole/l gas	7.92	7.67	0.065	0.06
CH_4_, mmole/g IVDDM	3.63	3.45	0.037	0.02
Total VFA, mM	37.1	36.6	0.451	0.64
Acetate (A), %	65.5	62.8	0.408	0.004
Propionate (P), %	23.2	25.2	0.379	0.006
Iso-butyrate, %	1.32	1.07	0.137	0.360
Butyrate, (%)	7.68	8.17	0.174	0.346
Iso-valerate, %	1.14	1.34	0.044	0.02
Valerate, %	1.20	1.43	0.039	0.002
A/P ratio	2.84	2.52	0.057	0.002
Microbial populations				
*Entodiniomorpha*, 10^4^/mL	2.79	2.30	0.096	0.007
*Holotricha*, 10^4^/mL	0.30	0.28	0.017	0.483
Total protozoa, 10^4^/mL	3.09	2.57	0.110	0.01
Total bacteria, 10^8^/mL	3.53	3.37	0.091	0.389
*Archaea*, 10^7^/mL	2.98	2.11	0.202	0.029

DM: dry matter; IVDMD: in vitro dry matter degradability; IVDDM: in vitro degraded dry matter; SEM: standard error of the mean.

**Table 4 animals-11-02843-t004:** The effect of *Paulownia* leaves with medium concentration of phenolic compounds (PLM), *Paulownia* leaves lamina (PLLA), and *Paulownia* twigs (PLT) on in vitro ruminal fermentation, methane production, and microbial populations (experiment 2).

Item	PLM	PLLA	PLT	SEM	*p*-Value
pH	6.54 ^b^	6.57 ^ab^	6.61 ^a^	0.008	<0.001
IVDMD	0.65 ^a^	0.65 ^a^	0.55 ^b^	0.009	<0.001
Ammonia, mM	11.2 ^a^	12.0 ^a^	9.2 ^b^	0.252	<0.001
Total gas, ml/g DM	304 ^a^	296 ^b^	276 ^c^	1.930	<0.001
CH_4_, mmole	1.02 ^a^	0.86 ^b^	0.76 ^c^	0.018	<0.001
CH_4_, mmole/g DM	2.55 ^a^	2.14 ^b^	1.90 ^c^	0.046	<0.001
CH_4_, mmole/l gas	8.37 ^a^	7.23 ^b^	6.84 ^c^	0.114	<0.001
CH_4_, mmole/g IVDDM	3.91 ^a^	3.34 ^b^	3.48 ^b^	0.056	<0.001
Total VFA, mM	38.9 ^b^	42.1 ^a^	34.9 ^c^	0.638	<0.001
Acetate (A), %	66.8 ^ab^	65.6 ^b^	67.6 ^a^	0.354	0.049
Propionate (P), %	21.8 ^b^	22.9 ^a^	20.5 ^c^	0.206	<0.001
Iso-butyrate, %	0.72	0.40	0.56	0.001	0.35
Butyrate, %	8.13	8.21	8.27	0.130	0.346
Iso-valerate, %	1.13 ^b^	1.47 ^a^	0.87 ^c^	0.054	<0.001
Valerate, %	1.17 ^b^	1.37 ^a^	0.89 ^c^	0.042	<0.001
A/P ratio	3.08 ^b^	2.87 ^c^	3.36 ^a^	0.047	<0.001
Microbial populations					
*Entodiniomorpha*, 10^4^/mL	2.09	1.82	1.92	0.099	0.235
*Holotricha*, 10^4^/mL	0.43	0.49	0.41	0.025	0.278
Total protozoa, 10^4^/mL	2.59	2.31	2.33	0.095	0.264
Total bacteria, 10^8^/mL	2.86	3.37	4.32	0.260	0.098
*Archaea*, 10^7^/mL	2.57 ^b^	1.86 ^c^	4.42 ^a^	0.440	0.035

DM: dry matter; IVDMD: in vitro dry matter degradability; IVDDM: in vitro degraded dry matter; SEM: standard error of the mean. Within each row, means with lower case superscripts (a–c) are significantly different at *p* < 0.05.

## Data Availability

The data presented in this study are available on request from the corresponding author.

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
