# Peer review of "The Effect of Different Concentrations of Total Polyphenols from *Paulownia* Hybrid Leaves on Ruminal Fermentation, Methane Production and Microorganisms"

_animals, 2021, doi:10.3390/ani11102843_

Round 1
Reviewer 1 Report
Title: The effect of different concentration of total polyphenols from Paulownia hybrid leaves on ruminal fermentation, methane production and microorganisms
The article proposes to evaluate the use of Paulownia as a ruminal fermentation manipulator, focusing on methane production. The manuscript is well written and contains relevant information on the topic. Some doubts arose during the reading of the material, which are presented below.
Line 68: tree descendent or originate from Chine?
Line 69: replace valuable dietary ingredient with potential dietary ingredient
Line 90 – 92: The authors characterized the soil of the places where the harvests were carried out. The doubt of this reviewer would be if only the soil could affect the composition of the material? Factors such as weather, stress due to lack or excess of water, temperature, etc... could affect the concentrations of the components? I think it would be important for the authors to describe the climatic conditions of the different places where the harvests were carried out.
Line 94 to 97: Regarding the concentrations used, are there any reasons to justify the choices? Are there any experiments that have been predicted to demonstrate that such doses of total polyphenols are passive to alter the ruminal fermentation process, particularly methanogenesis? I believe this information is important for the manuscript.
Line 150-151: please insert reference for NFC calculation.
line 168 to 170: why did the authors choose to conduct the first culture set evaluating PLH and PLL and in the second PLM, PLT and PLLA? Why don't they evaluate PLH, PLL and PLM, and then PLT and PLLA? This question needs to be very well answered by the data that shows that the doses used did not first study important statistical differences. These two numbers were replicated in the second experiment, using an intermediate dose.
line 219 to 226: in the statistical description, when "experiment 1 and 2" is described, do the authors refer to the two batch 168 culture sets? If so, I think it's important to standardize the writing, and make this very clear in the methodology.
Line 229 ate 235: The authors will carry out a statistical analysis differentiating Chemical composition of Paulownia. Would it be possible to apply a statistical analysis also to the contents of the main bioactive compounds?
Discussion: in the experiments conducted, could the results obtained be attributed to the basic nutrient composition of Paulownia, or the presence of polyphenols? Does the design of the experiment allow for this distinction? The authors need to make this more evident in the discussion of the material.
Line 291 – 302: In this excerpt, this reviewer had an important doubt. The difference in protein was attributed to the Paulownia varieties. Does the material collect at different locations sound of different varieties? I couldn't understand this clearly. Please rewrite this sentence.
In the discussion the authors first discussed the basic composition of Paulownia (DM, CP, NFC) and then discussed the concentration of phenolic compounds. Why did the discussion not follow the sequence of tables? If you are going to keep this sequence, please invert the sequence of Tables 1 and 2, as well as the presentation of the results.
Author Response
Dear Reviewer,
We would like to thank you for all of your comments and corrections.
We arranged our changes as follows:
- changed words/added or properly ordered, red fonts were used
- the bold text in this letter was used for the comments of Reviewers
-AU: stands for our replies, answers and comments
Below, please find our responses, point-by-point comments and suggestions.
Reviewer reports:
Reviewer #1:
The article proposes to evaluate the use of Paulownia as a ruminal fermentation manipulator, focusing on methane production. The manuscript is well written and contains relevant information on the topic. Some doubts arose during the reading of the material, which are presented below.
Line 68: tree descendent or originate from Chine?
AU: We introduced changes according to Reviewer suggestions. The tree originates from China.
Line 69: replace valuable dietary ingredient with potential dietary ingredient
AU: We introduced changes according to Reviewer suggestion.
Line 90 – 92: The authors characterized the soil of the places where the harvests were carried out. The doubt of this reviewer would be if only the soil could affect the composition of the material? Factors such as weather, stress due to lack or excess of water, temperature, etc... could affect the concentrations of the components? I think it would be important for the authors to describe the climatic conditions of the different places where the harvests were carried out.
AU: Required information regarding average temperatures and rainfall have now been described. However, no significant differences in these parameters were observed during the period of 3 months before harvesting.
Line 94 to 97: Regarding the concentrations used, are there any reasons to justify the choices? Are there any experiments that have been predicted to demonstrate that such doses of total polyphenols are passive to alter the ruminal fermentation process, particularly methanogenesis? I believe this information is important for the manuscript.
AU: Our previous study on Paulownia tomentosa and Paulownia fortunei hybrid proved that polyphenols mitigate rumen methanogenesis through a decrease in methanogens’ population (Huang et al., 2021). The content of polyphenols in that study was 24.6 mg/g DM and decreased methane production significantly. In the current manuscript we used Paulownia Clon In vitro 112 - Paulownia fortunei and Paulownia elongata hybrid. At the initial stage of the analysis of the polyphenols, it was found that the total concentrations of polyphenols differed among plantations, which prompted us to analyze the effect of different polyphenol contents on the methanogenesis process. The literature most often shows that higher concentration of total polyphenols affect the process of methanogenesis (> 40 g / kg diet; Mueller-Harvey 2006). It is well known that polyphenols exhibit antimicrobial, antiprotozoal and antibacterial properties by penetrating the cell membrane and disintegrating the structures of these microorganisms (Bodas et al., 2012). Tests on propolis extract containing phenolic acids (125, 250 or 500 μg propolis extract per 500 mg of DM, have shown a positive effect on the rumen processes (Morsy et al. 2013). Thus, our work aims to refine this knowledge. The design of the experiment was selected to show that different concentrations ofpolyphenols may have different effects on the rumen methanogenesis process (experiment 1). In the second in vitro experiment, different parts of PL were used.
We introduced the following sentence in line 177:
Line 150-151: please insert reference for NFC calculation.
AU: We introduced the reference according to your suggestions.
line 168 to 170: why did the authors choose to conduct the first culture set evaluating PLH and PLL and in the second PLM, PLT and PLLA? Why don't they evaluate PLH, PLL and PLM, and then PLT and PLLA? This question needs to be very well answered by the data that shows that the doses used did not first study important statistical differences. These two numbers were replicated in the second experiment, using an intermediate dose.
AU: We explained that situation above. The first experiment was conducted to evaluate the two extreme concentrations of total polyphenols on the responses of ruminal fermentation to examine concentration effect. The second experiment was conducted to study different leaves parts such as leaves with medium concentration of total polyphenols, lamella and twigs how parts can affect ruminal fermentation.
line 219 to 226: in the statistical description, when "experiment 1 and 2" is described, do the authors refer to the two batch 168 culture sets? If so, I think it's important to standardize the writing, and make this very clear in the methodology.
AU: Our research used: three repetitions and five replications (5 incubation vessels) for each of plant materials. Additionally, 5 bottles were added to each batch culture as the blank . The necessary clarifications have been added to the text of L 185 Finally, in first batch culture 35 bottles, in second batch culture 50 bottles were analyzed.
Line 229 ate 235: The authors will carry out a statistical analysis differentiating Chemical composition of Paulownia. Would it be possible to apply a statistical analysis also to the contents of the main bioactive compounds?
AU: Table 1 was supplemented with relevant calculations (now table 2). Information has also been added in the results section. L228-231All data were analyzed using SAS statistical software (Univ. Edition, version 9.4 SAS Institute Inc., Lane Cove, Australia). Chemical composition and contents of the main bioactive compounds of PL were tested by one-way analysis of variance (ANOVA), followed by a Tukey’s post-hoc test.L224-252
The higher content (P<0.001) of polyphenols was recorded in PLLA, PLH, and PLM (33.63, 31.35, and 26.94 mg/g DM, respectively), while the lowest concentration was noted in PLL and PLT (11.9, and 2.53 mg/g DM, respectively; Table 2). Among the different polyphenolic compounds, acteoside was the highest concentration in all of the analyzed samples of PL. Among the plant materials, the highest concentration of acetoside was noted in PLLA (21.15 mg/g DM) compared to others groups. Among all investigated PL samples, vast majority of biologically active substances (only except Apigenin 7) were significantly different (P<0.015). The 7- hydroxytomentoside, campneoside isomer and luteolin 4’,7-O-diglucuronide were dominant after acetoside.
Discussion: in the experiments conducted, could the results obtained be attributed to the basic nutrient composition of Paulownia, or the presence of polyphenols? Does the design of the experiment allow for this distinction? The authors need to make this more evident in the discussion of the material.
AU: We explaiened in the discussion part. Reduction of methanogenesis was obtained by lowering methanogens populations in PLH and PLLA whereas decreasing IVDMD caused a decreased efficiency in ruminal fermentation, mainly total VFA and reduced propionate proportion in PLT. Additionally, in PLT (with the lowest concentration of phenolic acids; 2.53 mg/g DM) a decrease in methane concentration was also observed (25 %), which was the resultant of increased NDF content in this part of leaves. Consequently, decreased methane production was a result of reduced digestibility of dry matter (by 15%) limiting hydrogen availability to methanogens. Increased NDF concentration induced a pH increase and IVDMD reduction with a direct impact on the content and VFA profile. Thereby, methane production reduction in the second experiment in PLLA and PLT groups was caused by different mechanisms. Moreover, PLT containing high fiber content decreased propionate proportion and acetate to propionate ratio, which are normally observed in high fiber forage diets compared with concentrate diets due to stimulation of fiber degrading bacteria that predominantly produce more acetate in comparison to propionate.
We hope that after the explanations above, the current version of the discussion is clearer.
Line 291 – 302: In this excerpt, this reviewer had an important doubt. The difference in protein was attributed to the Paulownia varieties. Does the material collect at different locations sound of different varieties? I couldn't understand this clearly. Please rewrite this sentence.
AU: Thank you for paying attention. The current version has been redrafted and we hope it will dispel the reviewer's doubts.
In the discussion the authors first discussed the basic composition of Paulownia (DM, CP, NFC) and then discussed the concentration of phenolic compounds. Why did the discussion not follow the sequence of tables? If you are going to keep this sequence, please invert the sequence of Tables 1 and 2, as well as the presentation of the results.
AU: The suggested changes were made in the results section.
We would like once again thank the Reviewer very much for all the valuable comments and suggestions that helped us to improve our manuscript.

Reviewer 2 Report
It is well written, so I have nothing major to point out.
My specific comments are as follows,
You show CH4/IVDMD, but I think In Vitro Digested Dry Matter, CH4/IVDDM is better.
L32; 112® --- You show ® only here. OK ?
L78; mg/g dry matter), --- mg/g dry matter(DM)),
L144; dry matter (DM), --- DM,
L197; L365; dry matter --- DM
L155; three ruminal cannulated Polish Holstein Friesian dairy cows --- Please show whether it has been reviewed by the animal ethics committee, permit number.
L183; (SRI310) --- Company name and place?
L184; Carboxen – 1000 column --- Company name and place?
L190; CP-104 --- Company name and place?
L204; Frisenette K02BP02500 --- Company name and place?
L205; Vacuport-Neuberg --- Company name and place?
L213; number H-1000 --- Company name and place?
L221; version 9.4 --- Company name and place?
L225; when <0.05. --- when P < 0.05.
L318; L336 --- You use H2, but others are hydrogen. Please unify.
L329; hydroksytomentoside --- English spell ?
L332; 7-Hydroksytomentoside --- 7-hydroksytomentoside
L347; Balcells et al. (2012), --- Please check this reference citation format and reference list.
Tables; Please see attached file written in red.

Author Response
Dear Reviewer,
We would like to thank you for all of your comments and corrections.
We arranged our changes as follows:
- changed words/added or properly ordered, red fonts were used
- the bold text in this letter was used for the comments of Reviewers
-AU: stands for our replies, answers and comments
Below, please find our responses, point-by-point comments and suggestions.
Reviewer reports:
Reviewer #2:
L32; 112® --- You show ® only here. OK ?
AU: We introduced changes according to Reviewer suggestions.
L78; mg/g dry matter), --- mg/g dry matter(DM)),
AU: We introduced changes according to Reviewer suggestions.
L144; dry matter (DM), --- DM,
AU: We introduced changes according to Reviewer suggestions.
L197; L365; dry matter --- DM
AU: We introduced changes according to Reviewer suggestions.
L155; three ruminal cannulated Polish Holstein Friesian dairy cows --- Please show whether it has
been reviewed by the animal ethics committee, permit number.
AU: We introduced the permit number.
L183; (SRI310) --- Company name and place?
AU: We introduced company and place.
L184; Carboxen – 1000 column --- Company name and place?
AU: We introduced Company name and place.
L190; CP-104 --- Company name and place?
AU: We introduced Company name and place.
L204; Frisenette K02BP02500 --- Company name and place?
AU: We introduced Company name and place.
L205; Vacuport-Neuberg --- Company name and place?
AU: We introduced Company name and place.
L213; number H-1000 --- Company name and place?
AU: We introduced Company name and place.
L221; version 9.4 --- Company name and place?
AU: We introduced name and place.
L225; when <0.05. --- when P < 0.05.
AU: We introduced changes according to Reviewer suggestions.
L318; L336 --- You use H2, but others are hydrogen. Please unify.
AU: We introduced changes according to Reviewer suggestions.
L329; hydroksytomentoside --- English spell ?
AU: We corrected into 7-hydroxytomentoside
L332; 7-Hydroksytomentoside --- 7-hydroksytomentoside
AU: We corrected it.
L347; Balcells et al. (2012), --- Please check this reference citation format and reference list.
AU: We corrected it.
Tables; Please see attached file written in red.
AU: Thank you for Reviewer suggestions. For methane we used in vitro degraded dry matter - IVDDM ((e.g., CH4/IVDDM) as you suggested, but for degradability we keep in vitro dry matter degradability - IVDMD. Please accept this.
We would like once again thank the Reviewer very much for all the valuable comments and suggestions that helped us to improve our manuscript.
Reviewer 3 Report
Dear authors,
Really thank you for this very very valuable piece of work. I truly believe that the quality and novelty of the paper are very good and surely of absolute Inrerest, given the time we are leaving and the challenge posed to livestock farming systems.
I enjoyed your manuscript very much. Congrats.
I have an argumentation (which is the real effect desired when you or we, as authors, write papers..)...what do you think would be the destiny of polyphenols in vivo, at those dosages you tested? And it is fascinating to see that poliphenols (naturally according to the different spectra and sources) once metabolized by GIT microflora are in part absorbed and fixed in tissues, in particular in bones. Supposedly, those could be used as dietary trackers, but also as origin trackers. Please, see Aldritt et al., 2019 Scientific Reports 9(1), 8047. Please, would you argument succinctly on this in your paper?
Thank you.
Author Response
Dear Reviewer,
We would like to thank you for all of your comments and corrections.
We arranged our changes as follows:
- changed words/added or properly ordered, red fonts were used
- the bold text in this letter was used for the comments of Reviewers
-AU: stands for our replies, answers and comments
Below, please find our responses, point-by-point comments and suggestions.
Reviewer #3:
Really thank you for this very very valuable piece of work. I truly believe that the quality and novelty of the paper are very good and surely of absolute Inrerest, given the time we are leaving and the challenge posed to livestock farming systems.
I enjoyed your manuscript very much. Congrats.
I have an argumentation (which is the real effect desired when you or we, as authors, write papers..)...what do you think would be the destiny of polyphenols in vivo, at those dosages you tested? And it is fascinating to see that poliphenols (naturally according to the different spectra and sources) once metabolized by GIT microflora are in part absorbed and fixed in tissues, in particular in bones. Supposedly, those could be used as dietary trackers, but also as origin trackers. Please, see Aldritt et al., 2019 Scientific Reports 9(1), 8047. Please, would you argument succinctly on this in your paper?
AU: Thank you, the Reviewer, for the kind words. This is a really interesting study that metabolites of polyphenol are deposited in the bone tissues and they may affect bone health and tissue formation. We conducted an in vivo experiment trying to determine the destiny of polyphenols. We did not take into account of the bones, only the content of individual polyphenols in the rumen fluid and milk. There are a few studies on the ffects of polyphenols on different tissues other than bone. Certainly, the study of Aldritt et al. will direct more research on the effects of polyphenol on bone health. Thanks again for suggesting this study, but we have no scope in this vitro study to discuss on the results of Aldritt et al.
We would like once again thank the Reviewer very much for all the valuable comments and suggestions that helped us to improve our manuscript.